# Selective Laser Melting and Spark Plasma Sintering: A Perspective on Functional Biomaterials

**DOI:** 10.3390/jfb14100521

**Published:** 2023-10-16

**Authors:** Ramin Rahmani, Sérgio Ivan Lopes, Konda Gokuldoss Prashanth

**Affiliations:** 1CiTin—Centro de Interface Tecnológico Industrial, 4970-786 Arcos de Valdevez, Portugal; sil@estg.ipvc.pt; 2proMetheus, Instituto Politécnico de Viana do Castelo (IPVC), 4900-347 Viana do Castelo, Portugal; 3ADiT-Lab, Instituto Politécnico de Viana do Castelo (IPVC), 4900-347 Viana do Castelo, Portugal; 4Department of Mechanical and Industrial Engineering, Tallinn University of Technology, 19086 Tallinn, Estonia; prashanth.konda@taltech.ee; 5CBCMT, School of Mechanical Engineering, Vellore Institute of Technology, Vellore 630014, Tamil Nadu, India

**Keywords:** functional biomaterials, porous lattice structures, laser powder bed fusion, tissue engineering, selective laser melting, spark plasma sintering

## Abstract

Achieving lightweight, high-strength, and biocompatible composites is a crucial objective in the field of tissue engineering. Intricate porous metallic structures, such as lattices, scaffolds, or triply periodic minimal surfaces (TPMSs), created via the selective laser melting (SLM) technique, are utilized as load-bearing matrices for filled ceramics. The primary metal alloys in this category are titanium-based Ti6Al4V and iron-based 316L, which can have either a uniform cell or a gradient structure. Well-known ceramics used in biomaterial applications include titanium dioxide (TiO_2_), zirconium dioxide (ZrO_2_), aluminum oxide (Al_2_O_3_), hydroxyapatite (HA), wollastonite (W), and tricalcium phosphate (TCP). To fill the structures fabricated by SLM, an appropriate ceramic is employed through the spark plasma sintering (SPS) method, making them suitable for in vitro or in vivo applications following minor post-processing. The combined SLM-SPS approach offers advantages, such as rapid design and prototyping, as well as assured densification and consolidation, although challenges persist in terms of large-scale structure and molding design. The individual or combined application of SLM and SPS processes can be implemented based on the specific requirements for fabricated sample size, shape complexity, densification, and mass productivity. This flexibility is a notable advantage offered by the combined processes of SLM and SPS. The present article provides an overview of metal–ceramic composites produced through SLM-SPS techniques. Mg-W-HA demonstrates promise for load-bearing biomedical applications, while Cu-TiO_2_-Ag exhibits potential for virucidal activities. Moreover, a functionally graded lattice (FGL) structure, either in radial or longitudinal directions, offers enhanced advantages by allowing adjustability and control over porosity, roughness, strength, and material proportions within the composite.

## 1. Introduction

The objective of tissue engineering is to create advanced biomaterials that are lightweight, possess high strength, and exhibit biocompatibility [1,2]. This goal arises from the increasing need for innovative composites to meet the demands of various applications in the biomedical field [3]. In recent years, intricate porous metallic structures, including lattices, scaffolds, and triply periodic minimal surfaces (TPMSs), fabricated using the selective laser melting (SLM) technique, have gained significant attention as potential load-bearing matrices for filled ceramics [4,5]. Within the biomaterial field, metal alloys, such as Ti-, Fe-, Co-Cr, Ta-Ti-, Ni-Ti-, and Mg-based alloys, are utilized in various forms, including porous structures or solid parts [6,7]. Additionally, these alloys can be combined with ceramic fillers to enhance their integration with bone tissues [8]. Conversely, copper-based alloys, when combined with silver, exhibit significant potential for antiviral and virucidal applications beyond live cells [9].

Among the ceramics commonly utilized in biomaterial applications are titanium dioxide (TiO_2_), zirconium dioxide (ZrO_2_), aluminum oxide (Al_2_O_3_), hydroxyapatite (HA), wollastonite (W), and tricalcium phosphate (TCP) [10,11]. These ceramics exhibit desirable properties, such as biocompatibility, corrosion resistance, bioactivity, osteoconductivity, strength, hardness, and fracture toughness, that make them suitable for various biological applications. To create metal–ceramic composites, the SLMed structures are filled with a compatible ceramic material using the spark plasma sintering (SPS) process [12,13]. This process enhances the mechanical properties and bioactivity of the composites, enabling their utilization both inside and outside living organism applications after undergoing minor post-processing steps [14].

The novel combined laser powder bed fusion and powder metallurgy (LPBF-PM) approach offers distinct advantages, including rapid prototyping and assured consolidation of the composite materials. However, despite these benefits, challenges persist in terms of optimizing the structure and molding design of the composites to achieve the desired functional characteristics [15,16]. Addressing these challenges is essential for advancing the manufacturing processes and enhancing the performance of metal–ceramic composites [17,18]. Depending on the specific needs for the sample size of fabrications, shape complexity, densification and porosity level, and number of productions, the SLM—as a LPBF subdivision—and SPS—as a PM prong—processes can be applied individually or in combination to achieve desired outcomes [19,20]. For higher technology readiness levels (TRLs) and shifting from Industry 4.0 to 5.0, collaborative synergy between humans and AI-driven machines is emphasized for improved productivity, creativity, and advanced capabilities in additive manufacturing (AM) and LPBF, with evolving precision and compatibility considerations [21,22].

This article aims to provide an overview of the fabrication of biocompatible metal–ceramic composites through the integration of SLM and SPS techniques. By exploring the current state of research and developments in this field, the present review will shed light on the progress made and the potential of these advanced composite materials in artificial bone implants and virucidal disinfectant areas. Additionally, this review will discuss the key challenges faced in the production of metal–ceramic composites and the strategies employed to overcome them.

## 2. Materials

Metals and ceramics play the main roles in the realm of biomaterial applications, where their unique properties make them indispensable for various biomedical purposes. Metal alloys, including titanium (CP-Ti and Ti6Al4V), iron (316L), cobalt–chromium (CoCr28Mo6), tantalum–titanium (Ta-Ti), nickel–titanium (Ni-Ti), and magnesium-based (Mg-Zn-Zr and Mg-Al-Zn-Mn) alloys, have garnered significant attention in orthopedics, tissue engineering, and dentistry due to their excellent mechanical strength, biocompatibility, and corrosion resistance [23,24,25,26]. Meanwhile, other metal-based alloys, such as nickel (In625, In718, and In939) and copper (Cu-Ni-Si-Cr, Cu-Cr-Zr, Cu-Sn, and Cu-Ni-Sn), are applied in the chemical industry and antiviral investigations [27,28,29,30]. These materials are often used in implants, prosthetics, and dental fixtures, providing enhanced durability and long-term performance within the human body. On the other hand, ceramics like hydroxyapatite (HA), wollastonite (W), titanium dioxide (TiO_2_), and zirconium dioxide (ZrO_2_) have found application in bone grafts, dental coatings, and tissue engineering scaffolds [31,32,33,34,35,36]. These ceramic materials boast exceptional bioactivity, mimicking the mineral composition of bone, promoting osseointegration, and contribute to controlled drug/nutrient delivery and facilitate tissue regeneration. These materials are shown in Figure 1. Apart from metals and ceramics, synthetic and biodegradable polymers are widely used in biomaterial engineering due to their versatility and biocompatibility. Examples include polyethylene, polyurethane, polylactide, polycaprolactone, and silicone. They are often used in 3D/4D printing of medical devices, drug delivery systems, and tissue engineering scaffolds [37,38].

### 2.1. Metals

Several metals and metal alloys have been utilized for tissue engineering and bone regeneration due to their biocompatibility, mechanical strength, and corrosion resistance. Some commonly used ones are Ti-, Fe-, Co-Cr-, Ni-Ta-, Ta-, and Mg-based alloys. It is worth noting that the choice of metal or alloy depends on factors like the specific application, load-bearing requirements, and desired degradation characteristics. Ongoing research aims to develop new materials and surface modifications to enhance the performance and biocompatibility of metals in tissue engineering and bone regeneration [39,40]. Titanium and its alloys, such as Ti-6Al-4V and Ti-6Al-7Nb, are widely employed in orthopedic and dental implants [41,42,43,44]. They possess excellent biocompatibility, low density, and high mechanical strength, allowing for osseointegration and long-term stability. Stainless steel, particularly 316L, has been used in orthopedic implants. It offers good mechanical properties and corrosion resistance; besides, it is relatively inexpensive compared to other materials [45,46]. Co-Cr-Mo alloys (such as CoCr28Mo6) are commonly used in orthopedics, especially for load-bearing implants, such as hip and knee replacements. They exhibit excellent mechanical strength, biocompatibility, and wear resistance [47]. Magnesium and certain magnesium alloys have attracted attention in recent years due to their biodegradable properties [48,49]. They can gradually degrade in the body, eliminating the need for implant removal surgeries. However, challenges related to corrosion and biocompatibility still need to be addressed, and 3D printing of porous structures of such alloys is in progress [50]. Tantalum has been used in orthopedic applications, particularly for bone implants and coatings. It exhibits good biocompatibility and corrosion resistance and has a low modulus of elasticity, resembling natural bone properties. Recently, Ta-Ti alloys have gained interest for the SLM process due to their biocompatibility, mechanical properties, lightweight nature, thermal characteristics, and their applicability in diverse fields, such as aerospace and biomedical engineering [51]. Finally, nitinol (nickel–titanium) is a shape memory alloy composed of nickel and titanium in an almost equal proportion. It finds applications in orthopedic devices, such as bone plates and fracture fixation devices. Specifically, it possesses superelasticity and shape memory properties, allowing for flexibility and adaptive response to mechanical stress [52]. 

### 2.2. Ceramics

#### 2.2.1. TiO_2_ and ZrO_2_


TiO_2_ is a common industrial chemical that is used in a wide range of products, including paints, coatings, cosmetics, and food additives. The toxicity of TiO_2_ has been extensively studied, and it is considered significantly non-toxic [53]. In addition, the toxicity is highly dependent on size, shape, and surface chemistry, as well as the type of cells exposed to it [54,55]. In terms of in vivo applications, the toxicity of TiO_2_ is also dependent on several factors, including the dose, the route of exposure, and the duration of exposure. Generally, TiO_2_ particles that are ingested orally are not absorbed into the body and are excreted in the feces, indicating a low toxicity. However, inhalation of TiO_2_ particles can lead to respiratory tract irritation and inflammation. Overall, the available evidence suggests that TiO_2_ is safe for use in the vast majority of industrial and consumer applications, but caution should be exercised when using TiO_2_ nanoparticles in certain in vitro and in vivo applications [56,57]. Using TiO_2_ and ZrO_2_ ceramics is recommended for in vitro applications, while alternatives like HA and W are better suited for in vivo applications [58,59]. When incorporating these oxides into lattice structures, it is important to investigate the adhesion at the interface of metal–ceramic boundaries [60,61]. The focus is on employing intricate porous metallic architectures, such as lattices, scaffolds, or TPMSs, which are fabricated through the LPBF method. These structures are intended to serve as robust load-bearing frameworks for the incorporated ceramics [62].

TiO_2_ and ZrO_2_ are both metal oxides that are widely used in various applications, including in the biomedical field. While both materials are generally considered safe, there are some differences in their toxicity profiles. TiO_2_ has been extensively studied and is generally considered to be a low-toxicity material. However, some studies have suggested that certain forms of TiO_2_, such as ultrafine particles, may have toxic effects, particularly on the respiratory system. These effects are thought to be due to the ability of TiO_2_ particles to generate reactive oxygen species (ROS) and cause oxidative stress in cells [63]. Owing to its antiviral properties outside of living organisms, it is utilized for creating self-cleaning surfaces in combination with a silver treatment [64]. ZrO_2_, on the other hand, is generally considered to be a biocompatible and low-toxicity material. It has been shown to be well-tolerated by cells and tissues and has been used in various biomedical applications, including as a dental implant material. However, some studies have suggested that certain forms of ZrO_2_, such as nano-sized particles, may have toxic effects, particularly on the lungs and immune system. These effects are also thought to be due to the ability of ZrO_2_ particles to generate ROS and cause oxidative stress in cells [65]. Kandel et al.’s [66] research unveiled the biocompatibility of synthesized TiO_2_-ZrO_2_, demonstrating bone matrix deposition. Furthermore, the developed bioactive material exhibited inherent antibacterial properties and excellent osseointegration capabilities. In another study, hierarchical TiO_2_-ZrO_2_ nanocomposite scaffolds were employed for tissue engineering purposes, specifically in the context of cancellous bone regeneration. The optimal sample, distinguished by its structural integrity, comprised 13% ZrO_2_ and underwent sintering at 550 °C for 2 h, according to Mahtabian et al. [67]. This study highlighted the significant enhancements in mechanical properties achieved by incorporating ZrO_2_ into the TiO_2_ scaffold for bone strength. Tiainen et al. [68] illustrated that adding 1% of ZrO_2_ resulted in a remarkable 16% increase in strength while maintaining porosity within the range of 89 to 93%.

The choice of which material is better among TiO_2_ and ZrO_2_ depends on the specific requirements, applications, and intended use, e.g., biocompatibility, chemical stability, mechanical strength and brittleness, conductivity importance, and optical properties [69]. In terms of biocompatibility, both TiO_2_ and ZrO_2_ are generally considered safe for in vivo and in vitro applications, although more attention has been paid to ZrO_2_ in the literature. Both exhibit good chemical stability; however, ZrO_2_ is known for its exceptional resistance to corrosion and chemical degradation, making it suitable for long-term implantation or exposure to harsh environments. ZrO_2_ is a significantly stronger and tougher material compared to TiO_2_. It has higher fracture toughness and can withstand mechanical stress and loading more effectively. This property makes ZrO_2_ desirable for applications where mechanical strength is crucial, such as dental implants or load-bearing orthopedic devices. TiO_2_ is widely used as a white pigment due to its high refractive index and excellent light-scattering properties. In contrast, ZrO_2_ is often employed as a transparent or translucent material, especially in dental restorations and optical applications [70]. Ultimately, ZrO_2_ exhibits higher electrical conductivity than TiO_2_, which can be advantageous in certain applications, such as sensors, fuel cells, or electronic devices, where electrical properties are essential.

For in vivo applications, where biocompatibility and mechanical strength are vital, ZrO_2_ may be a better choice. However, for in vitro applications that require optical properties or electrical conductivity, TiO_2_ might be more suitable. It is crucial to evaluate the specific needs and constraints of your application to determine which material is better suited for the intended purpose.

#### 2.2.2. Hydroxyapatite and Wollastonite

Both hydroxyapatite (HA, Ca_5_(PO_4_)_3_(OH)) and wollastonite (W, CaSiO_3_) are biocompatible ceramics that have been widely studied for their potential use in biomedical applications. Hydroxyapatite is a naturally occurring mineral and is the main inorganic component of bone. It is well known for its excellent biocompatibility and bioactivity, which make it an attractive material for bone tissue engineering and other biomedical applications. When used in implants or coatings, HA can enhance bone growth and improve osseointegration [71]. Note that osseointegration is a surgical technique that improves the quality of life and mobility of amputees by directly connecting a metal implant to the remaining bone of the limb, eliminating issues associated with traditional socket-based prosthetics and providing enhanced functionality and comfort. Wollastonite is also a biocompatible ceramic that has been studied for use in biomedical applications, particularly as a bone substitute. It has a similar chemical composition to bone and can be resorbed by the body over time. W also has good mechanical properties, making it suitable for load-bearing bone defects [72]. Both HA and W have shown good biocompatibility; however, the specific performance and biocompatibility of each material can vary depending on factors such as their microstructure, porosity, and surface characteristics. 

In addition to titanium dioxide, hydroxyapatite, and wollastonite, several other ceramics can be utilized for tissue engineering and bone regeneration purposes. Bioactive glasses (BGs), such as 45S5 Bioglass, can bond with living tissues and promote bone growth. They release ions that stimulate osteogenesis and exhibit excellent antibacterial effects and biocompatibility [73,74]. The creation of Bioglass 45S5 was a groundbreaking achievement, as it became one of the first artificial materials capable of chemically bonding with bone, leading to the development of other bioactive glass formulations. Its outstanding biocompatibility, preventing immune reactions and fibrous encapsulation, makes it highly suitable for various medical applications. Apart from hydroxyapatite, other calcium phosphate ceramics, like tricalcium phosphate (TCP) and biphasic calcium phosphate (BCP), are commonly employed. They possess osteoconductive properties and can be engineered to degrade at a controlled rate, allowing for new bone formation [75,76]. Aluminum oxide, also known as alumina (Al_2_O_3_), is another ceramic material with excellent mechanical properties and biocompatibility. It has been utilized in orthopedic implants and has demonstrated good wear resistance. The biodegradable Mg-HA cermet/cement shows promise as a permanent implant for bone regeneration, as an orthopedic implant, and in prosthesis applications [77,78,79]. The HA filler will be incorporated into an SLMed metallic matrix of Mg, formed using SPS technology, with the potential to benefit from LPBF’s rapid prototyping capabilities and cost-efficient methods of PM like hot isostatic pressing (HIP). 

## 3. Methods

Nowadays, functional biomaterials (FBMs) play a crucial role in advancing tissue engineering and regenerative medicine. The present study focuses on the fabrication of functionally graded lattice (FGL) structures and scaffolds using SLM and SPS methods. The integration of SLM and SPS techniques with these materials offers promising avenues for designing and producing highly tailored biomaterials with enhanced structural integrity and bioactive properties, paving the way for innovative applications in biomedical research, as shown in Figure 2. These FBM structures employ FGL metal alloys and bioceramic powders as the base materials, known for their biocompatibility and mechanical properties, showcasing their potential for combination and applications in tissue engineering.

### 3.1. Selective Laser Melting Process

SLM is an AM technique used to create 3D objects by selectively melting layers of metal powder using a high-powered laser. The SLM process offers several advantages, including the ability to produce complex geometries, intricate internal structures, and parts with high accuracy. It allows for the direct manufacturing of metal parts without the need for traditional machining (subtractive manufacturing (SM)) or casting processes (e.g., metal injection molding (MIM)), reducing material waste and enabling faster production times. It is a form of 3D printing that enables the production of complex and intricate metal parts with high precision and involves the following steps. The SLM process is schematically illustrated in Figure 3.

Design—The first step is to create a digital 3D model of the desired object using CAD software. The model is sliced into thin layers, typically ranging from 20 to 100 µm in thickness. These sliced layers serve as instructions for the SLM machine during the manufacturing process.

Powder preparation—Metal powders are selected based on the desired material properties and characteristics of the final object. The powder particles are typically gas atomized and spherical in shape and have a specific size distribution, mostly ranging from 10 to 60 µm, to ensure proper flowability and packing. The metal powder is then spread in a thin layer over the build platform.

Layer-by-layer melting—The build platform is lowered by the thickness of one layer, and a high-powered laser beam is directed onto the metal powder layer based on the instructions from the sliced 3D model. The laser selectively melts and fuses the metal powder particles together according to the cross-section of the current layer. The laser energy causes the powder to rapidly reach its melting point, allowing for fusion and solidification [80].

Cooling and solidification—Once a layer is melted and fused, the build platform is lowered, and a new layer of metal powder is spread on top. The process is repeated, layer by layer, until the entire object is created. As each layer cools, the molten metal solidifies, forming a solid part. The build platform is gradually lowered as new layers are added to compensate for the growing height of the object.

Post-processing—After the printing process is complete, the object is typically removed from the build platform and undergoes post-processing steps. This may involve removing support structures if they were used during printing, cleaning the object to remove excess powder, and performing any necessary heat treatment or surface finishing processes to achieve the desired final properties.

#### SLM of Biomaterial Fabrication

Understanding the mechanical properties of SLMed biomaterials is crucial for the development of safe and reliable medical devices and implants. By considering fatigue life during material selection, design optimization, and testing, engineers can enhance the performance and longevity of biomaterials in biomedical applications [81]. The design of a biomaterial component or device can impact its fatigue life. Factors such as stress concentration points, geometric features, corrosion resistance, and surface finish can influence the initiation and propagation of fatigue cracks. The magnitude, frequency, and type of cyclic loading applied to the biomaterial can affect its fatigue life. Higher stress amplitudes, increased loading frequencies, and alternating loads can decrease the fatigue life of the material. The fatigue behavior (as macroscopic properties) of porous biomaterials (as metamaterials) fabricated by SLM is significantly influenced by both the type of unit cell and porosity. In a study illustrated by Yavari et al. [82], it was demonstrated that biomaterials with a cube unit cell exhibited the longest fatigue life, followed by those with truncated cuboctahedron and diamond unit cells [83]. This research highlights the importance of considering the specific unit cell design and porosity when assessing the fatigue performance of porous biomaterials.

In the past decade, there has been considerable research interest in the development of porous bioactive materials using the SLM process. These materials have the ability to form a direct bond with living bone. Depending on their intended applications, these materials can have longitudinal or circumferential pores. One notable study by Fukuda et al. [84] focused on a structure featuring longitudinal square channels to determine the optimal conditions for promoting osteoinduction, which was observed to be most pronounced with the smallest tested diagonal widths, specifically 500 µm. In addition to bone regeneration, another important consideration is maintaining a balance between blood circulation and fluid movement within the material. Osteoconduction refers to the ability of a material or scaffold to provide a physical framework or structure that allows the migration of cells and the deposition of new bone. Osteoinductive materials can recruit and activate stem cells, triggering them to differentiate into bone-forming cells and promote the formation of new bone tissue. While both osteoconduction and osteoinduction are important mechanisms in bone regeneration, they involve different processes. Osteoconduction focuses on providing a physical scaffold for bone growth, while osteoinduction involves the chemical and biological signaling that induces the differentiation of stem cells.

Yavari et al. [85] reported a study on the application of as-manufactured Ti6Al4V for heat treatment and anodizing processes with the aim of enhancing its bioactivity and transforming the crystal structure of TiO_2_ nanotubes from anatase to rutile. The anodizing voltage was maintained at a constant value of 20 V for a duration of 60 min. The hierarchical surface transition from anatase to rutile began at 600 °C and was completed after 2 h. In another study, a process monitoring system utilizing an in-line photodiode was implemented to assess the structural integrity of titanium alloys [86]. This research aimed to decrease the laser input energy, increase the number of impacted layers, and subsequently reduce the load-bearing capacity. The study highlights the significance of establishing a correlation between the load-bearing capacity of porous structures and in situ process monitoring data.

Successful FBM fabrication using the SLM process involves a careful balance of laser energy, scan speed, optimal melting, and heat distribution, as well as microstructure and morphology survey and thorough evaluation of residual stresses. It is a multidisciplinary approach that combines materials science, engineering, and process optimization to achieve the desired biomaterial properties and performance. SLM enables control of relative density (porosity), surface quality (microstructure), and mechanical properties (microhardness) in Ti6Al4V alloys through varied laser energy inputs [87]. Exploring the scanning strategies in the SLM process to achieve homogeneous heating up to the melting point, considering the trade-off between power and scan speed, and comparing the results with numerical simulations is crucial [88]. J. Jhabvala et al. demonstrated that lower scan speeds result in cracks due to thermal gradients, while higher speeds lead to balling and unmolten powders at low laser settings. These experimental approaches were applied to low-conductive WC-Fe and high-conductive gold alloys. Residual stress evaluation using XRD analysis and FEM simulation in SLMed structures (316L and Ti6Al4V) is crucial to prevent crack formation, part deformation, and detachment from supports or substrates, both during and after processing, affecting both inner structure and surface morphology [89].

### 3.2. Spark Plasma Sintering Process

SPS is a relatively new and specialized powder metallurgy technique used for consolidating or sintering materials, typically ceramics, metals, and composites, into dense and high-quality components. It is a relatively rapid and energy-efficient process that offers advantages over traditional sintering methods [90,91,92]. The key advantages of SPS include its ability to achieve high densification levels, reduce sintering time, and retain the properties of the starting materials. It is particularly useful for materials that are difficult to sinter using conventional methods, such as those with high melting points or low diffusivity. The SPS process involves the following steps.

Powder preparation—The starting materials are typically in the form of fine powders. These powders can be pure metals, metal alloys, ceramics, composite mixtures, or powder filled/embedded into a structure. The powders’ size and shape are effective in the final product.

Mold assembly—The powders are placed in a die cavity or mold, which is made of a conductive material, such as graphite, or refractory metals, like tungsten or molybdenum. The mold is designed to accommodate the desired shape and size of the final component.

Pressure application—Pressure is applied to the powder compact using a hydraulic or mechanical press. The pressure helps to remove trapped gases, improve particle rearrangement, and enhance the density of the final product. The pressure can vary depending on the materials being sintered but is typically in the range of 10–200 MPa. For certain applications, this can manifest as a pressureless procedure or involve a rapid and pronounced fluctuation in pressure application. Furthermore, the SPS process commonly takes place within a vacuum chamber. 

Heating and spark plasma—Once the pressure is applied (preliminary or program-based), the assembly is subjected to intense pulsed electric currents. The electric current is passed through the mold, which acts as a resistive heating element. This causes the temperature of the powder compact to rise rapidly. During the heating process, electric sparks occur between adjacent powder particles due to the high electrical current passing through the mold. These sparks generate localized heating and facilitate the sintering process. The spark plasma phenomenon promotes rapid atomic diffusion and enhanced material transport, resulting in accelerated densification.

Sintering and cooling—The combination of pressure, heating, and spark plasma promotes the densification of the powder particles through plastic deformation, diffusion, and recrystallization. The material reaches a high temperature, typically below its melting point, allowing for particle bonding. Once the desired sintering temperature is achieved, the assembly is rapidly cooled, solidifying the component into its final form.

Figure 4 illustrates the schematic representation of the SPS process applied to FGL and FBM structures. These structures are intended to be filled with a specified ceramic ratio. The ceramic volume that is filled increases from the bottom to the top, while the lattice volume fraction is established throughout the SLM process.

#### SPS of Biomaterial Fabrication

The most-found medical metal materials in the market are Fe-, Co-, and Ti-based alloys. Numerous studies have demonstrated that incorporating pores into these materials can effectively decrease the Young’s modulus, thereby improving the compatibility between metal implants and human bones [93]. Among these materials, Ti-based alloys are particularly favored for implantable biomaterials due to their biocompatibility and lightness, unlike medical stainless steel (such as 316L austenitic) and Co-based alloys (such as Co-Cr alloy), which may contain toxic elements, like Ni, Al, V, and others. Titanium is stronger, lighter than iron, and has a lower elastic modulus and higher corrosion resistance. Regardless of the choice of materials and alloying techniques, the SPS method offers a fast, reliable, highly densified, and easily controllable approach to produce a wide range of biometals, ceramics, cermets, and glasses. Young’s modulus, stress shielding, and corrosion are critical factors that can be regulated through the optimization of temperature, pressure, and time during the SPS process [94]. Wu et al. [95] prepared gas-atomized Ti_45_Zr_10_Cu_31_Pd_10_Sn_4_, which is a Ni- and Be-free Ti alloy (non-toxic), and showed the loading pressure and density and porosity relationship. This material has high metallic glass-forming ability and biocompatibility and is suitable for both SPS and SLM processes. One of the older works on an SPSed Ti-Al-V-based material (Ti-6Al-4V, Ti-Al-V-Cr, and Ti-Al-V-Cr-Mn alloys) compared the surface morphology of this material with human osteoblastic cells [96]. The outcome of this article focusses on nanostructured/nanocrystalline titanium alloy biomaterials for implant acceptance, with tailored porosity, which is important for cell adhesion, growth, viability, and differentiation. Human osteoblast-like cells (MG-63) have been applied for cultivation on sintered porous Ti-based surfaces. MG-63 serves as a widely employed osteoblastic model for investigating the viability, adhesion, and proliferation of bone cells on load-bearing biomaterials, including titanium [97].

Different considerations are required when utilizing either SLM or SPS individually for fabricating porous structures without combining AM and PM approaches. In the SLM process, important factors include the CAD design of samples, material printability, and powder flowability. Conversely, in SPS, significant aspects involve die design for each specific sample shape and pressure control to achieve porous structures. Zhang et al. [98] demonstrated the preparation of pressureless porous Ti, with high porosity and pore sizes ranging from 50 to 500 µm, using a modified graphite die and temperatures ranging from 1000 to 1200 °C. It is worth noting that a scaffold created through SLM featuring circular and uniformly sized pores with a diameter of 400 μm and a porosity level of 35% exhibits a density of approximately 1.4 g/cm^3^ and a compressive strength of 110 MPa for silicon–wollastonite [99]. However, the strength of such SLM-built bioceramics remains uncertain and under development in the field of tissue engineering. According to the Gibson–Ashby model, porosity plays a critical role in determining the mechanical compatibility of porous biomaterials. Essential mechanical properties, such as elastic modulus and compression strength, are key parameters for the design of porous implants [100]. Although Ti6Al4V is the most widely recognized titanium alloy for studying biocompatibility in SLM, other titanium alloys, such as TiNi and Ti22Al25Nb, are also being considered for this purpose. Similarly, when it comes to the SPS process, there is ongoing research exploring different alloys. For instance, an investigation using the pressureless SPS process was conducted on the Ti5Al2.5Fe alloy, specifically targeting its application in hip joints and dental implants. The study revealed that the compressive strength of the porous samples was dependent on the sintering temperature, which ranged between 750 and 850 °C [101].

### 3.3. Effect of Powder in SLM and SPS 

The use of gas-atomized and pre-alloyed powders in sphere-shaped form significantly influences the effectiveness of AM, specifically in approaches like LPBF and SLM [102]. As-received powders often undergo vacuum or air drying to eliminate moisture, which can lead to issues related to flowability, porosity formation, and increased oxidation, particularly in alloys like AlSi10Mg containing substances such as MgO, Al_2_O_3_, and SiO_2_. Compared to Ti6Al4V and Inconel 718, AlSi10Mg is more susceptible to moisture and oxygen absorption, resulting in reduced spreadability and relative density on the build platform [103]. Conversely, when combining commercial metals, such as CP-Ti or Ti6Al4V, with hard ceramics, like TiB_2_ or coated diamonds, it is essential to mill them for an optimal duration before employing the SLM process. For instance, a mixture of pure Ti powder and 5 wt.% TiB_2_ composite exhibits more spherical particles after 2 h of milling, resulting in a 5% higher fabrication density following the SLM approach compared to a 4 h powder milling duration [104]. It is important to note that increasing the milling time has a negative impact on the porosity level, as revealed by this investigation.

The impact of powder morphology in the SPS process is not as significant as in the SLM process, unless specific physical properties, such as structural anisotropy, are explicitly required [105]. SPS commonly employs wide-shaped, e.g., nano-sized, flake-like, and gas-atomized, particles. When dealing with irregular shapes, large or elongated samples, or when combining SLMed porous structures with ceramic fillings inside lattices, the design of SPS molds for complex structures becomes crucial. The primary goals are achieving fully dense samples and controlled interface deformation [106]. In general, areas with greater thickness variations in the shape of a sample lead to more pronounced densification inhomogeneity during the SPS process. Design challenges often involve managing thermal gradients, differences in metal–ceramic melting points, unmolten powders from the SLM process, and mold–sample separation. Additionally, the simultaneous fabrication of multiple complex parts can pose a further challenge in ultrarapid sintering processes [107]. This capability, however, proves valuable for creating customized implants, prosthetics, or parts embedded within each other, where separation is facilitated using graphite sheets/foils. These graphite sheets serve as membranes/discriminants between the mold wall and samples, as well as between multiple parts. Innovations in molding design and the optimization of SPS parameters, including time, pressure, temperature, and heating rate, are of the utmost importance, especially when considering biomedical applications [108]. A ceramic composite mixture of CaSiO_3_/TiO_2_/HA exhibits higher hardness and compressive strength when sintered at 1250 °C compared to 1150 °C. To prevent damage to lattice/scaffold structures in metallic–ceramic composites, it is recommended to use a pressureless approach with a high heating rate. A successful example of this is the rapid-heated pressureless SPSing of ZrO_2_, utilizing a heating rate of 500 °C/min, a dwell time of 2 min, and sintering at 1600 °C, resulting in crack-free, homogeneous, and efficiently processed ceramics [109]. 

## 4. Functionally Graded Lattice

The preceding section elucidated the essential factors within SLM and SPS procedures. The initial heating, temperature elevation, dwell time during holding, and swift cooling stages in the SPS process are illustrated in Figure 5A. Additionally, Figure 5B delineates the scanning strategy (laser current, exposure time, and point distance) and defines the hatch spacing (including overlapping and the boundary counter). The scan strategy defines the pattern in which the laser scans the powder bed. Common strategies include raster scanning, parallel scanning, and contour scanning. The choice of scan strategy can influence the thermal gradients, residual stresses, and microstructure of the fabricated part. It is typically determined based on the part geometry, orientation, and desired properties [110]. In addition, the layer thickness determines the height of each deposited layer. It affects the resolution, surface roughness, and build time of the part. Thinner layers provide higher resolution but increase processing time, while thicker layers can reduce resolution but expedite the process. The layer thickness should be optimized based on the specific requirements of the application, powder characters, and metal conductivity [111,112].

SEM (scanning electron microscopy), TEM (transmission electron microscopy), XRD (X-ray diffraction), and EDS (energy-dispersive spectroscopy) represent the most-recognized microscopy techniques for characterizing composite structures. Typically, this stage occurs after processing (such as LPBF or PM) and post-processing (including procedures like polishing, powder jet cleaning, subtractive manufacturing, etc.) and precedes virucidal, biocompatibility, and mechanical testing. The schematic of this phase is displayed in Figure 5C. SEM and TEM provide imaging capabilities with varying depths of focus and high-resolution, three-dimensional images of the sample’s surface, whereas XRD reveals information about crystal structures and EDS identifies elemental composition, which is an analytical technique often used in conjunction with SEM or TEM. 

Plaques are clearings in a layer of cells caused by the infection and replication of viruses. Plaque-forming units (PFU) are a unit of measurement used in virology to quantify the number of viral particles capable of forming plaques in a viral culture or assay [113]. In virological studies, a PFU assay is commonly used to assess the infectivity and viral replication capacity of a virus and determine the viral titer in a sample. Colony-forming units (CFU) are used to quantify viable microorganisms, such as bacteria or fungi, in a sample. The CFU assay involves plating a diluted sample onto a solid agar medium, allowing individual viable cells to grow and form visible colonies [114]. CFU is commonly used in microbiology for environmental monitoring and food safety. Generally, CFU is used to measure viable bacteria by counting the number of visible colonies on solid agar in microbiology, while PFU is used to measure viable viral particles by counting the number of plaques formed during viral infection in virology. Viral culture is a laboratory test to find viruses that able to infect, which is shown in Figure 5D. Viral titer is determined by the number of plaques formed, to determine the strength of a virus against the host cells. For viral titer calculation (VTC) as PFU, it is needed to count the number of viral particles in a given volume of a sample. First, dilute the sample to obtain a countable number of viral particles and then plate the diluted sample onto a suitable host cell culture. After incubation for a specific time, you count the number of viral plaques or colonies that have formed, and based on the dilution factor, you can determine the viral titer expressed as PFU per milliliter or CFU per milliliter (PFU/mL or CFU/mL) [115,116]. 

The future of lattice, scaffold, TPMS, and 3D-printed porous structures is promising, with several potential developments on the horizon.

Improved resolution and accuracy—LPBF, and more specifically SLM technology, is constantly improving, with advances in laser and material technology allowing for the fabrication of structures with higher resolution and accuracy. This could lead to the creation of lattice structures with even greater complexity and specificity, allowing for more precise control over properties, such as porosity, cell design, compressive strength, and mechanical strength [117].

Development of new materials—The range of materials that can be used in SLM is expanding, with the development of new biocompatible and bioresorbable materials that could be used to fabricate lattice structures. These new materials could offer improved biocompatibility and better integration with surrounding tissues, leading to improved patient outcomes.

Integration with other technologies—SLM could be combined with other technologies, such as bioprinting or tissue engineering, to create even more complex structures. For example, lattice structures could be combined with living cells or tissues to create hybrid structures that have the potential to regenerate damaged or diseased tissues; or, as we suggested in the present article, they can be involved in PM processes to create bio-metal–ceramics. These composites are made by biocompatible ceramics to fill scaffold structures created through powder metallurgy for tissue engineering applications due to their excellent biocompatibility, bioactivity, osteoconductive properties, and innovative treatments for a variety of tissue defects and injuries.

Also, designing and modeling lattice and scaffold structures during the SLM process for biomedical applications requires careful consideration of several factors. 

Biocompatibility and biomechanical strength—The material used to create the lattice or scaffold structure should be biocompatible and non-toxic, with no potential for adverse reactions when implanted in the body. In addition, it needs to have favorable strength. For example, a scaffold designed for bone regeneration should be able to withstand compressive forces, while a scaffold designed for cartilage regeneration should be able to withstand shear forces.

Porosity and geometry complexity—The porosity of the structure should be carefully controlled to ensure that it is suitable for the intended application. A scaffold with high porosity may be suitable for tissue engineering applications, while a scaffold with lower porosity may be better suited for load-bearing applications [118]. The geometry of the structure should be carefully designed to ensure that it is stable and can support the intended load. The design should also consider any potential stress concentrations that could lead to material failure, in addition to stress shielding, due to its crucial role in promoting optimal healing and growth [119].

Surface roughness—The surface roughness of the structure should be carefully controlled to ensure that it is suitable for cell attachment, nutrient facilitation, waste removal, and proliferation. A rougher surface may be more conducive to cell attachment and growth, while a smoother surface may be better suited for load-bearing applications [120].

Manufacturing constraints—The design of the structure should take into account any manufacturing constraints associated with the SLM process. For example, the structure may need to be designed with support structures to prevent deformation during the printing process.

In AM technology, the terms “lattice” and “scaffold” are often used to describe two different types of structures used to optimize the design and functionality of 3D-printed objects. Lattice structures refer to a type of internal or external geometric pattern that is repeated throughout the object. These patterns (also known as cells) consist of a network of interconnected struts forming a porous or honeycomb-like structure. Lattices are designed to provide specific mechanical properties, such as lightweight, high strength-to-weight ratios, and enhanced energy absorption capabilities. They are commonly used to reduce the weight of 3D-printed parts without compromising structural integrity. The porous nature of lattices can facilitate better airflow, heat dissipation, and fluid flow through the object, making them suitable for applications in aerospace, automotive, and biomedical industries. Scaffold structures, on the other hand, are primarily used in the context of bioprinting or tissue engineering. In this field, scaffold structures are designed to mimic the extracellular matrix found in living tissues. The scaffold acts as a temporary support structure that provides a framework for cells to grow and organize into FBMs. Scaffold structures in bioprinting are typically created using biocompatible materials, such as hydrogels or polymers, that can be safely implanted into the body. These structures are designed with precise internal geometries, including pore size, shape, and interconnectivity, to allow for cell migration, nutrient diffusion, and waste removal. Over time, as the cells populate and replace the scaffold, it degrades naturally, leaving behind a fully functional tissue [121].

There is a third type of structure, TPMS, which is not typically considered a scaffold in the context of AM or bioprinting. TPMS refers to a specific type of surface that has a minimal surface area among all surfaces that can be periodically repeated in three dimensions. These surfaces have unique mathematical properties and are often characterized by complex, interconnected patterns. TPMS structures have various applications in different fields, but they are not primarily used as scaffolds for tissue engineering or bioprinting. Instead, TPMS structures find applications in areas such as materials science, surface coating, architectural design, metamaterials, and mathematical research. Some examples of TPMS structures include gyroids, diamond surfaces, and Schwarz. In materials science, TPMS structures can be utilized to enhance the mechanical properties of materials, such as lightweight and high-strength composites. They can also be employed in surface coatings to improve adhesion, create unique textures, or control the flow of liquids. In architectural design, TPMS structures can be used to create aesthetically pleasing patterns or optimize the strength-to-weight ratio of structures.

FGLs made by AM are structures that exhibit a gradual variation in their properties and geometries. They are designed to have different characteristics, such as varying porosity, mechanical properties, or thermal conductivity, across different regions of the lattice. This gradient allows for customized performance and functionality in specific applications. The FGL structure is characterized by a seamless transition starting from pure metal at the base, progressing through a gradient of a metal–ceramic lattice with varying ratios (e.g., SLMed), and culminating in a 100% dense ceramic oxide in the top (e.g., SPSed) for intense reinforcement in a single direction. An advantageous feature is its adaptability for radial, axial, or mixed designs, allowing the customization of bioceramic and biometal proportions according to specific applications [122,123]. This definition is shown in Figure 6. Benefits encompass the customization of mechanical, thermal, or other properties to meet specific requirements; the arbitrary design of lattice parameters, such as strut thickness, layer thickness, density, volume fraction, or connectivity; different regions of the lattice being able to exhibit different properties; the minimization of weight while maintaining structural integrity; and allowing for functionalities like sensing, actuation, or energy storage. This allows for improved performance, such as optimized strength, stiffness, energy absorption, or heat transfer, in different parts of the lattice structure.

### 4.1. Applications of FGLs and FBMs

FGLs made by AM have a wide range of potential applications across various industries.

Biomedical engineering—FGLs can be applied in the field of biomedical engineering for the design and fabrication of customized implants, prosthetics, or tissue scaffolds. By incorporating a gradient in porosity, mechanical properties, or bioactive features, it is possible to create implants that promote better integration with surrounding tissues or provide personalized solutions.

Aerospace and automotive—FGLs can be used in lightweight structural components for aerospace and automotive applications. By tailoring the lattice properties, such as stiffness or energy absorption, to specific regions of the structure (especially internal segments), improved performance and weight reduction can be achieved [124].

Energy storage, conversion, and heat exchange—FGLs can be utilized in energy storage devices, such as batteries or fuel cells, to optimize their performance. By varying the lattice properties, it is possible to enhance the conductivity, capacity, or durability of these devices. By designing a gradient in thermal conductivity, the lattice can facilitate efficient heat transfer and thermal regulation in specific areas, leading to improved energy efficiency [125].

Acoustic and vibration damping—FGLs can be employed to control acoustic or vibration characteristics in structures. By varying the lattice parameters, such as density or stiffness, it is possible to create structures that attenuate or redirect sound waves or dampen vibrations [126].

Robotic and mechanical devices—FGLs can be integrated into robotic systems or mechanical components to enhance their performance. By tailoring the lattice properties to specific regions, it is possible to optimize strength, flexibility, or energy absorption, enabling improved functionality and durability. The versatility of additive manufacturing and the ability to customize lattice properties offer opportunities for innovation in various industries, e.g., tribology and tunnel boring machines (TBM), with weldability and hard material filled [127,128]. 

### 4.2. Pore and Strut Precision

Achieving precise pore morphology in porous biomaterials is crucial to ensure optimal performance. However, geometric and mechanical mismatches can arise, leading to issues such as pore occlusion and strut thinning, which can negatively impact bone ingrowth and compromise the mechanical integrity of the material. To address these challenges, a compensation strategy is proposed to minimize the mismatch between the as-designed and as-built geometry, particularly in relation to pore morphology. Focusing on the application of SLM, a compensation scheme based on an error analysis in a spider web model has been reported by Bagheri et al. [129]. By investigating the sources of geometric deviations and developing strategies to mitigate them, this study aimed to enhance the accuracy and quality of pores in biomaterials. To enhance design efficiency and reduce weight, a promising approach involves utilizing the combined power of SolidWorks-Ansys software (or other CAD-FEM coupling software, like Materialise, COMSOL, etc.) to simulate and optimize porous structures. Moreover, in the post-processing phase, the application of alumina nanoparticle jet cleaning can effectively eliminate adhered or unmelted particles. When considering porous biomaterials, such as lattices or scaffolds, the pivotal elements are the pores and struts. The accuracy of the pores is crucial for ensuring efficient delivery of oxygen or drugs [130]. In the design process, it is important to account for vertical and horizontal available surfaces for scaffolds, as well as the uniform or gradient diameter of pores for FGLs, tailored to specific applications. On the other hand, the struts play a crucial role in bearing loads and meeting the chemical and mechanical requirements of bone structures.

Strut and cell design also effect the fatigue life of structures. Hooreweder et al. conducted a study to enhance the durability of biomedical implants manufactured using SLM by employing a series of post-built surface and heat treatments. These treatments were found to be crucial in significantly improving the fatigue life of porous structures. The research focused on modifying the microstructure and minimizing stress concentrators and surface roughness to enhance the fatigue life of biomaterials. The effects of stress relieving, hot isostatic pressing, and chemical etching were thoroughly investigated in [131]. In addition to their biocompatibility, the porous structures exhibit favorable characteristics for structural components, such as enhanced strength-to-weight or stiffness-to-weight ratios. To evaluate their performance, the SLMed porous structures can be compared with stress-relieved samples obtained through heat treatment. Additionally, comparisons can be made with structures fabricated using other methods like LPBF combined with post-processing techniques, like SPS, HIP, or MIM. Furthermore, the performance of chemically etched surfaces can also be assessed as a surface treatment option. Drawing inspiration from metamaterials (engineered materials with intricate structures designed to exert control and manipulation over a wide range of physical phenomena, including light, sound, thermal energy, electromagnetic waves, etc.), the development of strut-based biological and geometric topology metamaterials offers a promising solution for achieving multiphysical performance regulation. In this context, the importance of the design of strut types is emphasized, and strut topological design is investigated to understand its impact on mass transport properties [132]. 

The Ti6Al4V alloy is widely recognized as a metallic alloy extensively used for implants due to its exceptional biocompatibility. While the Ti6Al4V bulk material possesses a Young’s modulus of approx. 110 GPa, the modulus of bone is approximately approx. 20 GPa. This significant difference in mechanical properties gives rise to the phenomenon of stress shielding, which negatively impacts the longevity of dental implants. Porosity plays a vital role in dental implants as it influences stiffness, which is closely linked to the Young’s modulus and plays a crucial role in reducing the effective modulus of porous metallic structures [133]. Essential factors contributing to porosity include unit cell size, strut diameter, pore size, volume fraction, and FGL structure, which can be controlled using as-designed CAD software and as-built SLM devices.

## 5. In Vivo, In Vitro, and In Silico Studies

This article presents a visionary perspective on the imperative need to integrate both SM and AM approaches to achieve rapid and highly efficient cermet fabrication in biomedical applications. By combining these two manufacturing approaches, we can unlock the full potential of cermet materials in addressing crucial biomedical issues. The utilization of subtractive manufacturing allows the precise shaping, solidification, densification, and refining of cermet structures, while additive manufacturing enables the creation of intricate designs with enhanced complexity and customization. This synergistic approach promises to revolutionize the biomedical field by enabling the production of advanced cermet components that exhibit superior innovative solutions, performance, and biocompatibility. In particular, our suggested model for the integration of powder bed fusion and powder metallurgy technologies facilitates the production of a versatile cermet composite tailored for a broad spectrum of in vivo and in vitro bioengineering applications.

It is worth mentioning that our preferred structure is a perpetual implant designed to remain permanently inside the body (as the in vivo goal). This implant consists of two main components: a functionally graded porous metallic structure and a ceramic part filled with a specified volume fraction in each desired section. This unique design serves various purposes, like drug delivery release, bone regeneration, and gradient reinforcement, longitudinally or radially. The next generation of bone regeneration holds great promise as it can be achieved through a combination of bioceramics, such as HA-W-TCP, along with materials possessing high strength-to-weight ratios and excellent impact damping capacity, such as Mg-Zn-Zr, Mg-Al-Zn-Mn, Mg-Nd-Zn-Zr, Mg-Nd-Zr-Y-Gd, Al-Si-Mg, and Al-Mg-Sc [134].

On the other hand, we have developed a successful model for in vitro testing of antiviral, antibacterial, and antiadhesive composite materials specifically designed for environments that encounter bacteria and viruses, such as laboratories [135]. To assess their effectiveness, the constitutive parts of these materials underwent testing for their ability to combat bacteriophages. One of the most well-known bacteriophages used in this testing is Phi6, a thoroughly characterized enveloped RNA virus that serves as an established model system for virucidal experiments. It has been utilized as a benchmark for studying eukaryotic viruses like Ebola and coronaviruses [136]. In addition to assessing the antiviral properties, we will also evaluate the antibacterial features of the composite materials. For this purpose, we will utilize various model bacteria species, including *E. coli*, *B. subtilis*, *C. jejuni*, and *S. aureus*, which will be fluorescently tagged for tracking purposes. These bacteria will be cultivated to different physiological states to examine the impact of the new materials on their adhesion. Preventing the transmission of bacteria within hospitals is crucial due to the abundance of surfaces that can harbor harmful microbes. Nosocomial infections pose a real threat, and incorporating antimicrobial touch surfaces can be an effective strategy against pathogenic bacteria. Among the various materials available, copper stands out as a popular choice, either in its pure form or alloyed with elements like Cu-Ni-Si-Cr, Cu-Cr-Zr, Cu-Sn, Cu-Al, and Cu-Ni-Sn, owing to its remarkable contact killing efficiency. Additionally, silver and certain oxides, such as TiO_2_, ZrO_2_, and Al_2_O_3_, also demonstrate self-cleaning and antimicrobial properties [137]. Given these characteristics, compositions like Cu-Ag-TiO_2_-ZrO_2_, or similar combinations, emerge as the top candidates for in vitro testing as they hold promise in combating bacterial transmission effectively.

When it comes to manufacturing metal–ceramic composites, the choice of additive manufacturing techniques plays a significant role. For porous samples, SLM proves to be favorable, while SPS is preferred for solid/bulk samples. However, the combination of LPBF-PM (as proposed in this article), referring to the merging of SLM and SPS, becomes inevitable when dealing with ceramic-embedded metals. Currently, ceramic 3D printing using LPBF methods is progressing based on different heat treatments, but concerns remain regarding compression strength, crack formation, and brittleness [138]. On the other hand, SM excels in creating solid metallic parts, but its limitations in achieving lightness and reducing stress shielding reduction drive us towards AM techniques. In this context, the LPBF-PM approach proves beneficial as it allows the deliberate incorporation of pores in the ceramic phase through SPS parameters and precise control of the volume fraction in the metal matrix via SLM parameters. Metal matrix composites (MMCs) find extensive use across various applications and are produced using a variety of techniques [139,140]. This enables us to evaluate the final production’s brittleness, ductility, lightness, porosity level, and proportion of components. These factors are essential for designing effective metal–ceramic composites, ensuring the correct volume fraction of each component.

The significance of cermet composites lies in the fact that ceramics are generally more brittle than metals, meaning that they tend to fracture or break easily under stress without significant plastic deformation. On the other hand, metals are typically more ductile, allowing them to undergo plastic deformation before failure, making them better suited for applications requiring toughness and elongation, and offer good strength-to-weight ratios. Ceramics generally have a higher specific strength and stiffness, making them lighter per unit volume compared to most metals. Figure 7 illustrates the integration of ceramics within a metallic framework, encompassing a variety of materials. This involves the utilization of the SLM technique for antiviral or biomaterial metals, like Cu or Mg, followed by the SPS process for incorporating ceramics with virucidal and biocompatible properties, such as TiO_2_-Ag or W-HA [115,141].

As an example, hip and femur replacements typically involve the use of artificial implants made from materials like stainless steel, cobalt–chromium alloys, ceramics, or a combination of materials [142,143]. These implants are designed to mimic the natural anatomy and function of the hip joint and femur, providing support, stability, and improved mobility for individuals with hip joint problems, such as arthritis, fractures, or other conditions. These materials are chosen for their biocompatibility, strength, durability, and resistance to corrosion. To overcome the problem of heaviness and stress shielding, porous structure, metal–ceramic composites, lighter metals, and AM approaches are suggested (see Figure 7).

Magnesium has a low density and is commonly known for its lightweight properties. It is lighter than materials like aluminum, making it a popular choice in various applications where weight reduction is important, such as in the automotive and aerospace industries. On the other hand, there are some problems for 3D printing of Mg by SLM. To overcome these challenges, ongoing research and development efforts are focused on improving the process parameters, developing specialized powder formulations, and optimizing the printing conditions. Mg alloys bearing the commercial designations ZK30/ZK60, JDBM, WE43, and AZ31/AZ61/AZ91 have garnered significant attention due to their remarkable potential in promoting osteogenesis imperfecta healing (for bone fractures and disorders), attributed to their osteoconductivity, biodegradability, and printability [144,145].

High reactivity—Magnesium is highly reactive and prone to oxidation. During the SLM process, exposure to oxygen and high temperatures can lead to oxidation and the formation of oxide layers on the printed part’s surface. This can result in poor mechanical properties and reduced dimensional accuracy.

Flammability and explosion risk—Magnesium is a flammable material, and the fine magnesium powder used in SLM printing poses a fire and explosion risk. Special precautions, such as strict control of atmosphere and powder handling procedures, are required to ensure safety during the printing process.

Thermal conductivity—Magnesium alloys have high thermal conductivity, which can result in rapid heat dissipation during the SLM process. This rapid cooling can cause thermal gradients and residual stresses in the printed part, leading to distortion, warping, or cracking.

Susceptibility to porosity—Magnesium alloys are prone to the formation of pores or voids during the SLM process. This can be attributed to the low boiling point of magnesium, which leads to the vaporization of the material under the laser’s heat. Proper parameter optimization, including laser power and scanning speed, is crucial to minimize porosity [146].

Limited powder availability—Compared to other commonly used metals in SLM, such as Al and Ti, the availability of magnesium powders suitable for SLM printing is relatively limited. This can restrict the material options and hinder the widespread adoption of magnesium alloy SLM printing.

The terms in vivo, in vitro, and in silico refer to experiments which, respectively, are performed in living organisms, inside laboratories, and as computer simulations. Combining computer-aided design (CAD) software, like SolidWorks, and finite-element method (FEM) software, like ANSYS, enables us to assess the complicated shaped gradient and uniform lattice structures before fabrication and in the in silico stage [122,124,147]. Testing virus stability, virucidal effect, virus inactivation, and bacteriophage infectivity in contact with metals, ceramics, and cermet composites can be performed under in vitro conditions to assess the contamination of a specimen’s surface and the virucidal potential of a composite. The delivery and release of antibiotics, vitamins, oxygen, or liquids depend on longitudinally, latitudinally, and radially graded pores. During consideration for an in vivo application, applying the FGL/FBM structure avoids the stress shielding phenomenon and enables us to fill the metallic lattice with the desired ceramic too [148,149]. Figure 8 depicts the context of these experiments within our study, with a particular emphasis on bone regeneration and the reduction in viral contamination using the LPBF-PM approach. This illustration outlines a vertical sequence within a rectangle, presenting appropriate metal alloys, objectives, traits, domains of application, and desirable ceramic compositions, arranged from the upper section to the lower section.

Cu and its alloys, including brass and bronze, have inherent antimicrobial properties and possess self-cleaning properties. They exhibit what is known as the “oligodynamic effect”, where they can kill or inhibit the growth of a wide range of microorganisms, including bacteria, viruses, and fungi. Meanwhile, there are some problems in SLM 3D printing of pure Cu and its alloys [150,151,152]. 

High reflectivity and thermal conductivity—Copper and its alloys have high reflectivity for laser beams, especially at the wavelength commonly used in SLM printing. The high reflectivity can lead to inefficient energy absorption and reduced laser–material interaction. This can result in poor melt pool formation, incomplete melting, and reduced part density. Cu alloys have exceptionally high thermal conductivity, which can lead to rapid heat dissipation during the SLM process. The high thermal conductivity can make it challenging to maintain a sufficiently high temperature in the melt pool. It can also cause heat transfer to the surrounding areas, resulting in thermal gradients and undesirable residual stresses.

Susceptibility to oxidation and material contamination—Copper alloys are prone to oxidation when exposed to air or elevated temperatures. During the SLM process, the high laser power and localized heating can cause oxidation of the copper powder, leading to the formation of oxide layers on the printed part’s surface. Oxidation can negatively affect the mechanical properties and dimensional accuracy of the printed parts. Cu alloys can be susceptible to contamination from other materials. This can occur during the SLM process due to residual powder from previous builds or cross-contamination between different material batches. Contamination can affect the alloy’s composition, resulting in variations in mechanical properties and compromising the integrity of the printed parts.

Challenges in post-processing—Cu and its alloys are often difficult to machine or process due to their high hardness and ductility. Post-processing steps, such as machining, polishing, or heat treatment, can be more challenging compared to other materials. These additional processing difficulties can increase the overall production time and costs. Addressing these challenges requires careful process optimization, including adjusting laser parameters, controlling the atmosphere during printing to minimize oxidation, and implementing effective powder handling and cleaning procedures to prevent contamination. Researchers and engineers are continually working to develop new techniques and methodologies to overcome these obstacles and for the SLM process through AM technology, and one of the main steps is using the green laser range. 

Currently, fabricating pure Cu using the conventional LPBF route poses a significant challenge [153]. The typical approach involves an infrared laser with a wavelength around 1000 nm or higher [154]. However, at this wavelength, Cu’s energy absorptivity decreases, and the high thermal conductivity (approximately 400 W/m-K) hinders proper formation of the molten pool. Some suggested methods to overcome these issues include controlling layer thickness, reducing powder size, increasing laser power, and adjusting process parameters [155,156]. Nevertheless, a more promising solution involves utilizing a laser source with a lower wavelength, such as a green laser with approximately 500 nm. By doing so, it becomes feasible to fabricate complex architectures with dense microstructures and achieve continuous plastic deformation at a higher strain rate [157]. Beyond its mechanical, electrical, and thermal properties, pure or alloyed copper in lattice or bulk structures exhibits antiviral and self-cleaning effects [158]. Notably, Cu-based alloys combined with Ag and TiO_2_ create a novel highly virucidal cermet composite [115].

In the context of testing self-cleaning surfaces of Cu-TiO_2_-Ag (in vitro study), viruses can be used as a model organism to assess the effectiveness of these surfaces in preventing viral contamination. Self-cleaning surfaces are designed to repel or inactivate microorganisms, including viruses, bacteria, and fungi, to maintain a cleaner and more hygienic environment [159]. When testing self-cleaning surfaces, researchers may use surrogate viruses that are harmless to humans but share similar characteristics to pathogenic viruses. Bacteriophages, which are viruses that infect bacteria, are often employed as model organisms in these studies. They can be chosen based on their structural similarities to specific viruses of interest or their ability to mimic the behavior of pathogens. These tests provide valuable information about the efficacy of self-cleaning surfaces in reducing viral contamination. There are several bacteria options that can be used for testing virus stability instead of Phi6 bacteriophages. These bacteria are commonly employed in virology research to assess the stability, survival, and behavior of viruses. A few alternative bacteria are Escherichia coli (*E. coli*), used for laboratory research and studying bacteriophages and other viruses [160]; Salmonella enterica (*S. enterica*), used for viral stability studies and frequently used as a model organism and in the study of various aspects of viral infection; Pseudomonas aeruginosa (*P. aeruginosa*), used as a versatile bacterium for microbiology research; Staphylococcus aureus (*S. aureus*), used for investigating the persistence and survival of viruses on surfaces; and Bacillus subtilis (*B. subtilis*), a Gram-positive bacterium which is frequently used as a model organism for various research purposes. 

## 6. Industrial Revolution

The transformation of industrial revolution from 4.0 to 5.0 signifies the shift from fully automated and connected systems to a new era of collaboration, where humans and AI-powered machines work hand in hand to achieve unprecedented levels of productivity, creativity, and problem solving [161]. As for Industry 5.0, which represents a future paradigm of manufacturing, its specific approaches toward lightweight AM and LPBF might not be fully defined yet, given that the concept is still evolving [162]. The evolution entails complete automation and process control spanning from 3D scanning, 3D designing, and 3D printing to post-processing and quality assessment of samples. Simultaneously, there are advancements in rapid prototyping devices and personalized powder technology [163]. As Industry 5.0 advances with the emergence of 4D printing, it promises enhanced efficiency in tissue engineering and orthopedic implants [164]. Notably, this calls for integrated software, like Scan-Design-Analysis and SolidWorks-Ansys, with seamless import–export capabilities [165,166]. A diverse array of cell designs is available for the fabrication of porous biomedical structures using LPBF/SLM. These designs can be categorized as strut-based, involving arrangements like body-centered cubic (BCC), face-centered cubic (FCC), body-centered cubic zero (BCCZ), or face-centered cubic zero (FCCZ) arrangements; surface-based arrangements, encompassing vertical, horizontal, or radial plate configurations; or volume-based arrangements, utilizing TPMS arrangements, such as Schwarz, Diamond, and Gyroid structures [167,168].

## 7. Mechanical Tests and Properties

Mechanical testing of porous structures for biomedical applications is crucial to ensure that these structures possess the required mechanical properties to support tissue growth, withstand physiological loads, and maintain structural integrity. The primary mechanical tests and properties are illustrated in Figure 9. The choice of mechanical tests will depend on the specific application and the intended tissue or organ being engineered. Several mechanical tests can be performed on these structures.

Compression test—Compression tests involve applying axial loads to the scaffold/lattice structure to evaluate its compressive strength, stiffness, and deformation behavior. Sing et al. [169] conducted a study to examine how process parameters affect dimensional accuracy and compressive behavior. They observed that as laser power or laser scan speed increased, the thickness of powder adhesion on the struts decreased. Additionally, they found that an increase in relative density led to a higher elastic constant in compression for the lattice structures. This is important for applications where the scaffold will experience compressive forces, such as in the femur, humerus, and other long bones.

Tensile test—Tensile tests assess the scaffold’s ability to withstand tensile forces. Tensile strength, elastic modulus, and strain-to-failure ratio are important parameters to determine. This test is particularly relevant for scaffolds used in soft tissue engineering. Scaffolds displayed a deformation behavior primarily characterized by stretching when subjected to tensile loads, and their stiffness and strength were influenced by the level of porosity [170]. Adjustments in laser parameters led to enhanced fatigue resistance in tensile loading, with a notable improvement observed, especially in the gyroid microarchitectural design.

Shear test—Shear testing evaluates the resistance of the scaffold to shear forces. It is important for scaffolds used in applications where shear loads are significant, such as cartilage or meniscus tissue engineering. Due to the inability of the body to naturally heal substantial bone defects, persistent endeavors are dedicated to advancing the field of 3D scaffolds for bone tissue engineering [171]. Zero-shear viscosity is a property of a material’s viscosity when it is unaffected by shear stress and holds significance in tissue engineering and rheology studies. It can be relevant when studying the flow behavior of biomaterials, such as hydrogels or other materials used for creating tissue scaffolds or drug delivery systems.

Biodegradation assessment—In some cases, mechanical testing may be combined with degradation studies to assess how the mechanical properties of the scaffold change over time as it degrades in the body. This is particularly relevant for biodegradable scaffolds [172].

Pore size (porosity) and strut diameter (accuracy)—While not a traditional mechanical test, analyzing the pore size distribution and porosity of the scaffold is essential for understanding its permeability and ability to facilitate nutrient and waste exchange within the tissue. Qui et al. [173] demonstrated that the laser scanning speed primarily affected strut thickness when operated at slower rates, with the highest porosity observed at intermediate speeds. Furthermore, high-speed imaging revealed that increasing laser power resulted in a larger melt pool.

Fatigue test—Fatigue tests are conducted by subjecting the scaffold to repeated cyclic loading to assess its resistance to fatigue failure. This is particularly important for scaffolds that will be subjected to repetitive loading in vivo, such as those used in joint tissue engineering. Clearly, the key factors influencing fatigue behavior are cell size and strut diameter. In a study conducted by Zhao et al. [174], various unit cell types (tetrahedron and octahedron) and pore sizes (500 µm and 1000 µm) were fabricated using the SLM process. The results revealed that octahedron scaffolds exhibited superior static mechanical properties, longer fatigue lives, and higher fatigue strength when compared to their tetrahedron counterparts. As anticipated, scaffolds with 1000 µm pores exhibited lower compressive properties and shorter fatigue lives when contrasted with those featuring 500 µm pores.

Fracture toughness test—This test assesses the scaffold’s resistance to crack propagation. This is relevant in situations where the structure may be exposed to potential sources of damage, e.g., HA, W, BCP, and TCP. An essential factor affecting fracture toughness is the building direction of metallic lattices during the SLM process. Alsalla et al. [175] discovered that the density of lattice structure samples remained consistent, regardless of whether they were built vertically or horizontally. However, it was observed that the samples built in the vertical direction exhibited superior tensile and fracture toughness properties when compared to those constructed in the horizontal direction.

Mechanical properties of lattice and scaffold structures should be tailored to match the specific requirements of the target tissue and the intended application. The structures used in biomedical applications should possess specific mechanical properties to fulfill their intended functions and support tissue regeneration. The desired mechanical properties can vary depending on the target tissue, organ, and application [176]. Some key mechanical properties that lattice and scaffold structures can have for biomedical applications are detailed here.

Load bearing strength—The scaffold should have sufficient strength to withstand the mechanical loads it will encounter in the body. The required strength varies widely depending on the tissue being replaced, with load-bearing tissues like bone requiring higher strength than soft tissues. Carluccio et al. [177] assessed the current state of biodegradable metal processing through SLM for load-bearing bone scaffold applications and conducted a meta-analysis to understand the impact of processing parameters on relative density. Synthetic bone scaffolds are gaining in popularity for treating critical bone defects, and SLM offers a means to create customized scaffolds with complex architectures. While the SLM process for biodegradable metal is still emerging, it is evident that future research should focus on broader guidelines for SLM machines to optimize the manufacturing efficiently.

Elastic modulus—The scaffold’s elastic modulus (stiffness) should match that of the surrounding tissue as closely as possible to avoid stress shielding or mechanical mismatch. This property is especially critical for orthopedic and musculoskeletal applications. Kadirgama et al. [178] conducted a study to investigate the correlation between various factors and the Young’s modulus of structures with values ranging from 0.01 to 1.84 GPa. Their findings highlighted the substantial influence of porosity in this relationship.

Porosity and pore size—Controlling the porosity and pore size of the scaffold is essential to promote cell infiltration, nutrient diffusion, and waste removal. The specific values depend on the tissue type and the desired level of porosity [179,180].

Biodegradability and degradation rate—Some scaffolds need to be biodegradable, gradually breaking down as new tissue forms. The rate of degradation should be tailored to match tissue regeneration rates [181]. Zinc exhibits low melting and boiling points, leading to increased porosity in the fabricated components during the SLM process. Demir et al. [182] explored the possibility of achieving greater porosity by manufacturing under varying atmospheric conditions. They introduced an innovative approach involving an open chamber setup with an inert gas jet flowing over the powder bed.

Interconnectivity—Interconnected pores and channels within the scaffold facilitate cell migration, nutrient exchange, and tissue integration. Proper interconnectivity is particularly crucial in complex tissues like vascular or neural tissue [183]. 

Surface roughness—Surface roughness can influence cell adhesion and proliferation. Controlled roughness can be beneficial for promoting cell–scaffold interactions [184].

Fracture toughness—For tissues subjected to potential impact or crack propagation, a high fracture toughness is essential to resist fracture or damage. In a research investigation focused on Ti6Al4V, Cain et al. [185] examined a comparison between heat-treated (post-processing) and as-built structures concerning fracture toughness and crack growth rate. The most significant enhancement in properties through heat treatment was notably observed when the fracture plane was perpendicular to the SLM build direction. This alteration during heat treatment plays a crucial role in achieving rapid densification and enhancing mechanical properties.

Thermal and electrical conductivity—In some specialized applications, thermal properties may be important, such as in scaffolds used for thermal ablation or hyperthermia treatments. Scaffolds with electrical conductivity may be required for applications involving electrical stimulation or integration with electronic devices, such as cardiac tissue engineering or neural interfaces. As demonstrated in a study conducted by Butler et al. [186], the processing parameters and scanning strategies employed in SLM process exert a substantial influence on both porosity and thermal conductivity. While there is a strong inclination toward additively manufactured components with minimal porosity, this research highlights the potential to significantly lower laser energy density requirements, by roughly one order of magnitude, while still achieving acceptable levels of thermal conductivity.

Viscoelasticity and anisotropy—In some applications, especially soft tissue engineering, viscoelastic properties become important as the scaffold must mimic the behavior of natural tissue under dynamic loads. In cases where tissue has anisotropic properties (e.g., muscle or tendons), scaffolds can be designed to exhibit anisotropic behavior by aligning the structural elements in specific directions [187].

In short, the mechanical properties of lattice structures in the SLM process are vital for tailoring them to specific tissue and application requirements. Load-bearing strength, elastic modulus, and fracture toughness are crucial considerations, with ongoing research exploring methods to optimize these properties [188,189,190]. Controlling porosity, pore size, interconnectivity, and surface roughness is essential for promoting cell integration and functionality. Additionally, biodegradability and degradation rates must align with tissue regeneration rates.

## 8. Summary

This article explores the synergy of laser powder bed fusion and powder metallurgy in creating functional biomaterials. The process involves crafting intricate metallic structures like lattices and scaffolds using selective laser melting (SLM). In a subsequent step, various ceramics, such as titanium dioxide, zirconium dioxide, hydroxyapatite, and wollastonite, are seamlessly integrated through spark plasma sintering (SPS). The advantages of this approach encompass the rapid prototyping of complex shapes, the creation of lightweight yet robust structures, tailored designs for both in vivo and in vitro applications, exceptional consolidation, and robustness in metal–ceramic composites. Additionally, the spotlight falls on promising metal alloys, including Cu-Ni-Si-Cr, Cu-Cr-Zr, Cu-Sn, and Cu-Ni-Sn alloys of copper, along with Mg-Zn-Zr, Mg-Al-Zn-Mn, Mg-Nd-Zn-Zr, and Mg-Nd-Zr-Y-Gd alloys of magnesium. These alloys, along with TiO_2_, ZrO_2_, Al_2_O_3_, HA, W, and TCP ceramics, are envisioned to play pivotal roles in production and process evolution. In the realm of in vivo applications, the precise configuration of pores and struts emerges as a critical factor for controlled drug delivery and mechanical integrity. Simultaneously, achieving biocompatibility and wear resistance takes precedence in the context of permanent implants for bone regeneration. On a divergent note, the domain of in vitro applications promises lightweight, visually appealing structures with inherent self-cleaning and virucidal properties. To surmount challenges associated with 3D printing of pre-alloyed Mg and/or Cu in the context of the Industry 4.0 and 5.0 revolutions, strategic focus should encompass the evolution of SLM devices, the standardization of post-processing procedures, and the development of robust biomedical and/or antiviral testing methodologies. This holistic approach underscores the potential for advancing the landscape of functional biomaterials. The main points and objectives of this review article are succinctly outlined as follows:-The combination of selective laser melting and spark plasma sintering for advanced biomedical manufacturing.-The process involves creating intricate metallic porous structures (lattices, scaffolds, and TPMSs) using laser powder bed fusion.-Incorporating ceramics, such as titanium dioxide and zirconium dioxide, is suitable for in vitro applications, whereas hydroxyapatite and wollastonite are better suited for in vivo use.-Benefits include rapid prototyping, lightweight yet strong designs, and tailored biomaterial applications by additive manufacturing of copper and magnesium alloys.-Promoting devices and software during the fourth and fifth industrial revolutions to shape the future of tissue engineering.

## Figures and Tables

**Figure 1 jfb-14-00521-f001:**
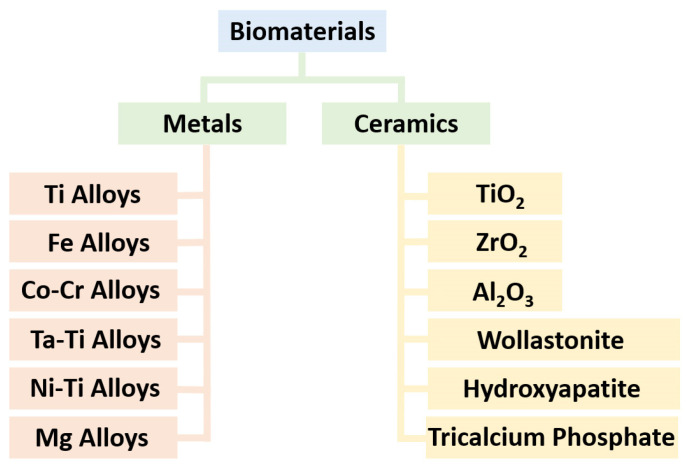
Integration of metals and ceramics in tissue engineering for the fabrication of artificial biomaterials.

**Figure 2 jfb-14-00521-f002:**
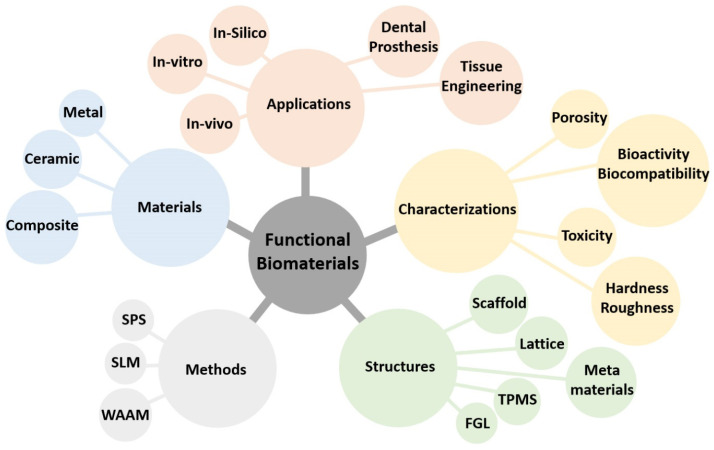
Schematic of functional biomaterial fabrication and applications using additive manufacturing.

**Figure 3 jfb-14-00521-f003:**
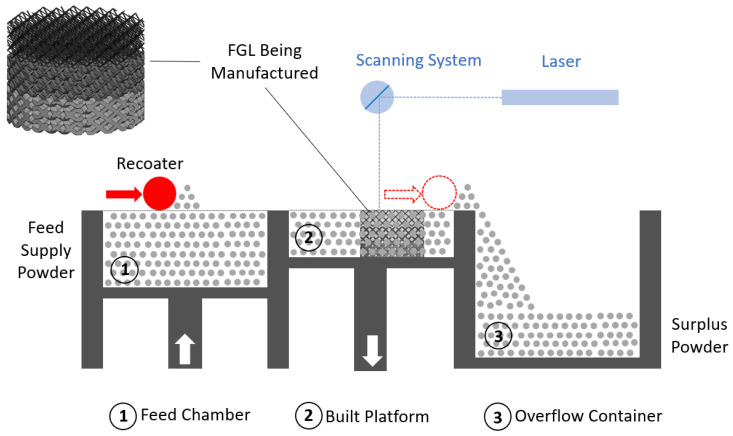
Schematic of the selective laser melting process.

**Figure 4 jfb-14-00521-f004:**
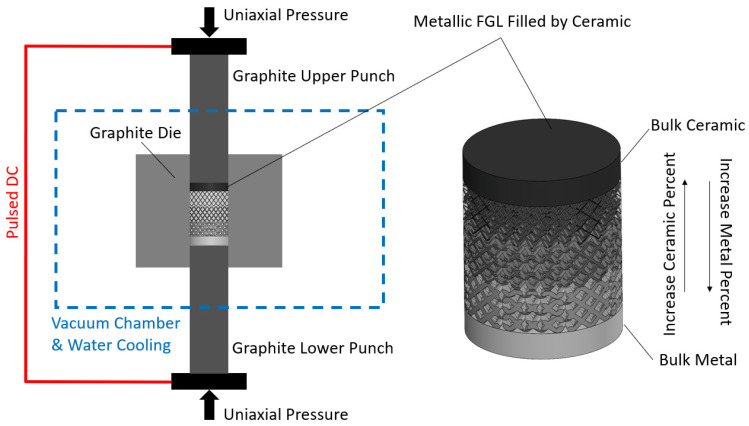
Schematic of the spark plasma sintering process.

**Figure 5 jfb-14-00521-f005:**
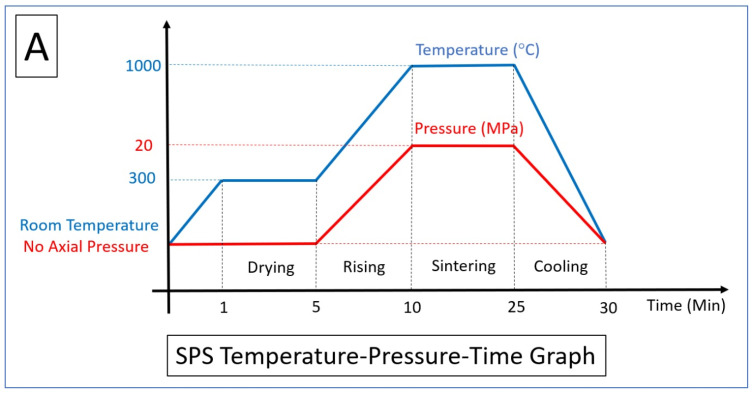
Key parameters for SPS–SLM processing and measurements: (**A**) SPS parameters, (**B**) SLM parameters, (**C**) Micrograph and characterization, and (**D**) Viral tests.

**Figure 6 jfb-14-00521-f006:**
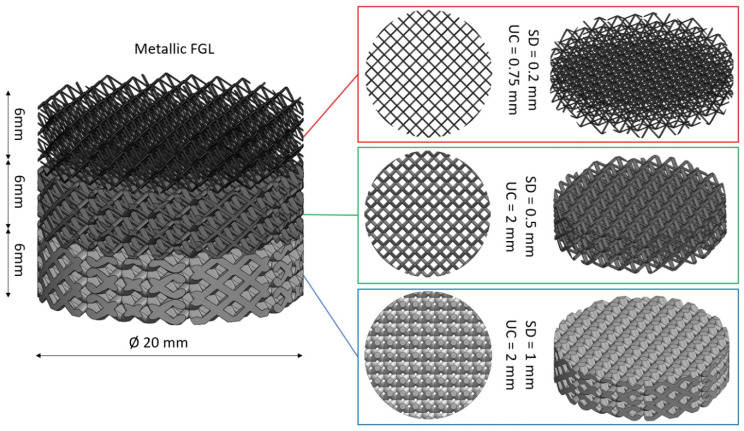
CAD design of a functionally graded lattice (FGL). All samples have a diameter of 20 mm and a height of 6 mm. Strut diameter (SD) and unit cell (UC) size, from top to bottom, are, respectively, (0.2, 0.75), (0.5, 2.0), and (1.0, 2.0) mm.

**Figure 7 jfb-14-00521-f007:**
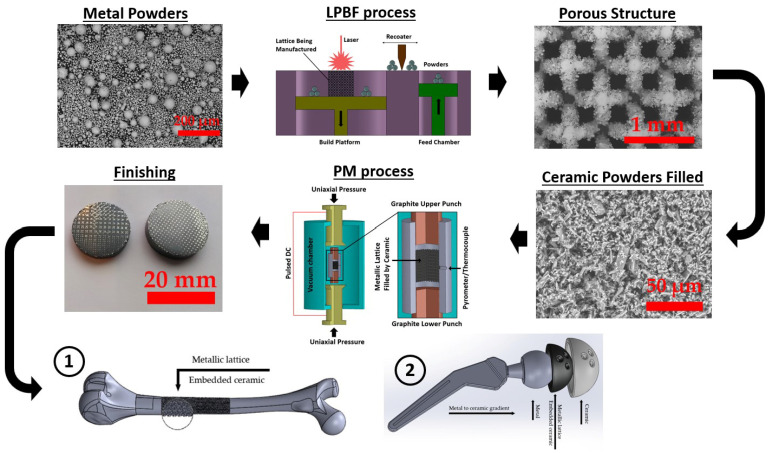
Fabrication of ceramic-doped metal through the LPBF-PM combined procedure. Note that the figure is not a representation of a certain process but rather a depiction of possible options for materials and methods. The figure shows copper-based Cu15Ni8Sn alloy, a schematic of the SLM process, a 316L lattice structure, wollastonite irregular-shaped powder, a schematic of the SPS process, and a Ti6Al4V lattice filled by a TiO_2_ oxide ceramic, respectively.

**Figure 8 jfb-14-00521-f008:**
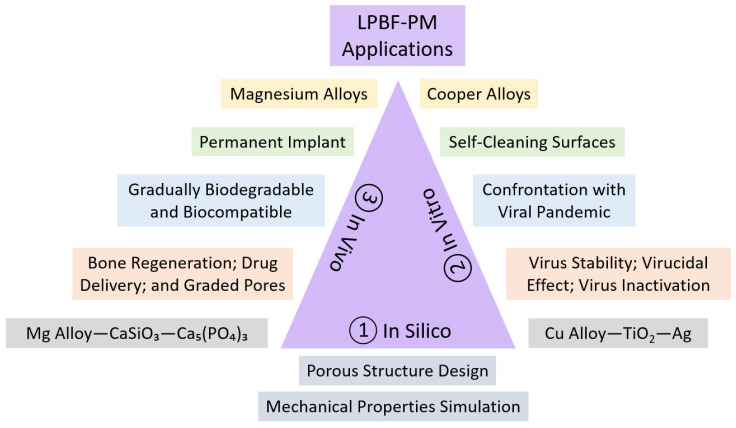
LPBF-PM solution for the regeneration of bone and reducing viral contamination.

**Figure 9 jfb-14-00521-f009:**
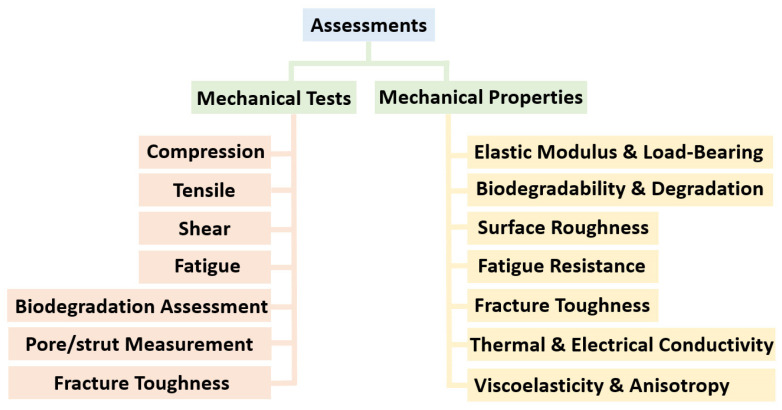
Mechanical tests and properties of porous structures for biomedical applications.

## Data Availability

Not applicable.

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
