# Peer review of "Selective Laser Melting and Spark Plasma Sintering: A Perspective on Functional Biomaterials"

_jfb, 2023, doi:10.3390/jfb14100521_

Round 1

Reviewer 1 Report

The authors wrote the manuscript on “Laser powder bed fusion and powder metallurgy: A
perspective on functional biomaterials”. The manuscript is written and discussed well.
However, the authors should add the below information before publishing:
1. All figures need to reference their source (even if from open access).
2. Cite the below related recent paper as:
https://doi.org/10.1016/j.ceramint.2022.10.071
3. Line 373 should be highlighted as subheading.
4. Line 533 and 534 should be alligned

none

Author Response

The authors would like to express their gratitude for your dedicated effort and time spent on reviewing our article. We have provided the responses to your comments and the revised manuscript for your perusal, which are attached herewith.

Reviewer 2 Report

Review Manuscript Title: “Laser Powder Bed Fusion and Powder Metallurgy: A Perspective on Functional Biomaterials”

1. This is an interesting manuscript that will be of interest to scientists working in the area of tissue engineering. The manuscript provides an overview of the fabrication of metal-ceramic composites using integrated Laser Powder Bed Fusion (LPBF) and Powder Metallurgy (PM) techniques. The manuscript describes advances in the field and strategies to address challenges associated with the production of metal-ceramic composites.

2. There are numerous abbreviations in the manuscript. The large number of abbreviations can be distracting. The authors should provide a list of abbreviations and the definition for the abbreviations at the beginning of the manuscript.

Overall, the quality of English language is excellent. There are minor errors that should be corrected.

Author Response

(The authors gave the same response as above.)

Reviewer 3 Report

Comments to authors are required to be addressed properly:

·         Authors should rephrase or clarify the phrase in lines 533 and 534. Also, the font is too large!

·         Some figures regarding the mechanical properties and applications should be included.

·         The conclusions

·         Authors should be cited with a reference the paragraph in lines 571-574. Also, authors are required to justify the reasons “a rougher surface  may be more conducive to cell attachment and growth, while a smoother surface may be  better suited for load-bearing applications.”

·         Authors did not show enough discussion and illustration regarding Figure 9.

·         The review paper lacks to present enough information and illustrations regarding the applications of porous metallic structures in biomedical applications.

·         The conclusions should concentrate on some significant points from the review paper and avoid general statements.  

English writing of this paper should be improved. 

Author Response

(The authors gave the same response as above.)

Reviewer 4 Report

In this Manuscript “Laser Powder Bed Fusion and Powder Metallurgy: A Perspective on Functional Biomaterials”, the authors explored the synergy of laser powder bed fusion (LPBF) and powder metallurgy (PM) in creating functional biomaterials. The process involves crafting intricate metallic structures like lattices and scaffolds using selective laser melting. In a subsequent step, different ceramics such as titanium dioxide, zirconium dioxide, hydroxyapatite, and wollastonite are seamlessly integrated through spark plasma sintering. The authors claimed that the combined SLM-SPS approach offers advantages such as rapid prototyping and assured consolidation, although challenges persist in terms of large-scale structure and molding design.

Overall, the research article has some shortcomings, which need to be addressed before possible publication in this journal.

Please find the attached annotated file to see my comments.

Journal of Functional Biomaterials publishes high-quality review articles related to biomaterials. Based on my comments, the recommendation is Major Revision.

Moderate editing of English language required.

Author Response

(The authors gave the same response as above.)

Reviewer 5 Report

The authors conducted a commendable review of SLM and SPM for functional biomaterials. However, the article requires a revision to enhance its coherence and readability, particularly in the alignment of sections.

1. Please include the list of the abbreviations. 

2. Please ensure the use of specific terminologies such as SLM and SPM instead of referring to more general techniques such as LPBF and PM. This is important because LPBF and PM encompass a wide range of styles, and the authors may need to add other techniques under these categories and compare them to demonstrate why the reviewed techniques are more relevant to 4.0 to 5.0. 

3. Line 27 - 28 (based on 2 pt). The authors started with SLM and SPM. However, the abstract closed with LPBF and PM. So to avoid confusion for the readers, it is suggested to revise the title and the lines. 

4. Also, please add a couple of lines about how the transformation of 4.0 to 5.0 will be advantageous based on the current review techniques in the abstract. It was stated in heading 4.0, but the abstract does not project anything related to this sub-title. 

5. Please revise the headings to avoid confusion. For instance, a section on SLMed biomaterials and section 2 materials (generalised). Thus, the subsection can be revised for based biomaterials to avoid confusion. Similar to SPM. 

6. Please revise the title headings 4. Industry 4.0 to Industry 5.0. Probably move this heading towards the end before the conclusions. 

7. Please revise the 4.2, 4.2.1, and 4.2.3 into a heading (4 or 5) Scaffolds produced using SPM and SLM. 

8. Heading 4.3 demonstrates in vitro and in vivo studies. These must be stand-alone heading (6): Biocompatibility of SLM/SPM-based biomaterials. In addition, please divide the sections in vitro (6.1), in vivo (6.2) and microbiology (6.3) etc., 

9. Including citations for statements based on research would significantly improve the quality of the information provided. It helps to establish credibility and allows readers to explore the topic further. Also, please check if the figures need citations and copyrights. 

Author Response

(The authors gave the same response as above.)

Round 2

Reviewer 3 Report

Comments to authors are listed.

·         The application and aims need to be stated clearly in the abstract section.

·         Mechanical Tests and Properties section still is required to enhance the information withy some numerical values  such as tables or figures to exhibited the section clearer.

The English writing of this paper should be improved. 

Author Response

Enclosed with this comment, you will find the cover letter addressing the reviewer's comments, along with the manuscript where relevant revisions have been highlighted in yellow.

Reviewer 4 Report

The article is in acceptable form, now.

Author Response

The authors express their gratitude for the acceptance of our article.